# Provable Membership Inference Privacy

**Zachary Izzo**
Department of Mathematics
Stanford University
zizzo@stanford.edu

**Jinsung Yoon**
Google Cloud AI Research
jinsungyoon@google.com

**Sercan O. Arik**
Google Cloud AI Research
soarik@google.com

**James Zou**
Department of Biomedical Data Science
Stanford University
jamesz@stanford.edu

## Abstract

In applications involving sensitive data, such as finance and healthcare, the necessity for preserving data privacy can be a significant barrier to machine learning model development. Differential privacy (DP) has emerged as one canonical standard for provable privacy. However, DP's strong theoretical guarantees often come at the cost of a large drop in its utility for machine learning; and DP guarantees themselves can be difficult to interpret. In this work, we propose a novel privacy notion, membership inference privacy (MIP), to address these challenges. We give a precise characterization of the relationship between MIP and DP, and show that MIP can be achieved using less randomness compared to the amount required for guaranteeing DP, leading to smaller drop in utility. MIP also guarantees are easily interpretable in terms of the success rate of membership inference attacks. Our theoretical results also give rise to a simple algorithm for guaranteeing MIP which can be used as a wrapper around any algorithm with a continuous output, including parametric model training.

## 1 Introduction

As the popularity and efficacy of machine learning (ML) have increased, the number of domains in which ML is applied has also expanded greatly. Some of these domains, such as finance or healthcare, have ML on sensitive data which cannot be publicly shared due to regulatory or ethical concerns (Assefa et al., 2020; Office for Civil Rights, 2002). In these instances, maintaining data privacy is of paramount importance and must be considered at every stage of the ML process, from model development to deployment. During development, even sharing data in-house while retaining the appropriate level of privacy can be a barrier to model development (Assefa et al., 2020). After deployment, the trained model itself can leak information about the training data if appropriate precautions are not taken (Shokri et al., 2017; Carlini et al., 2021a).

Differential privacy (DP) (Dwork et al., 2014) has emerged as the gold standard for provable privacy in the academic literature. Training methods for DP use randomized algorithms applied on databases of points, and DP stipulates that the algorithm's random output cannot change much depending on the presence or absence of one individual point in the database. These guarantees in turn give information theoretic protection against the maximum amount of information that an adversary can obtain about any particular sample in the dataset, regardless of that adversary's prior knowledge or computational power, making DP an attractive method for guaranteeing privacy. However, DP's strong theoretical guarantees often come at the cost of a large drop in utility for many algorithms (Jayaraman and Evans, 2019). In addition, DP guarantees themselves are difficult to interpret by non-experts. There is a

2022 Trustworthy and Socially Responsible Machine Learning (TSRML 2022) co-located with NeurIPS 2022.

precise definition for what it means for an algorithm to satisfy DP with $\varepsilon = 10$, but it is not a priori clear what this definition guarantees in terms of practical questions that a user could have, the most basic of which might be to ask whether or not an attacker can determine whether or not that user's information was included in the algorithm's input. These constitute challenges for adoption of DP in practice.

In this paper, we propose a novel privacy notion, membership inference privacy (MIP), to address these challenges. Membership inference measures privacy via a game played between the algorithm designer and an adversary or attacker. The adversary is presented with the algorithm's output and a "target" sample $\mathbf{x}^*$, which may or may not have been included in the algorithm's input set. The adversary's goal is to determine whether or not the target sample was included in the algorithm's input. If the adversary can succeed with probability much higher than random guessing, then the algorithm must be leaking information about its input. This measure of privacy is one of the simplest for the attacker; thus, provably protecting against it is a strong privacy guarantee. Furthermore, MIP is easily interpretable, as it is measured with respect to a simple quantity–namely, the maximum success rate of an attacker. In summary, *our contributions* are as follows:

- We propose a novel privacy notion, which we dub membership inference privacy (MIP).

- We characterize the relationship between MIP and differential privacy (DP). In particular, we show that DP is sufficient to certify MIP and quantify the correspondence.

- In addition, we demonstrate that in some cases, MIP can be certified using less randomness than that required for certifying DP. (In other words, while DP is sufficient for certifying MIP, it is not necessary.) This in turn generally means that MIP algorithms can have greater utility than those which implement DP.

- We introduce a "wrapper" method for turning any base algorithm with continuous output into an algorithm which satisfies MIP.

## 2   Related Work

**Privacy Attacks in ML**    The study of privacy attacks has recently gained popularity in the machine learning community as the importance of data privacy has become more apparent. In a *membership inference* attack (Shokri et al., 2017), an attacker is presented with a particular sample and the output of the algorithm to be attacked. The attacker's goal is to determine whether or not the presented sample was included in the training data or not. If the attacker can determine the membership of the sample with a probability significantly greater than random guessing, this indicates that the algorithm is leaking information about its training data. Obscuring whether or not a given individual belongs to the private dataset is the core promise of private data sharing, and the main reason that we focus on membership inference as the privacy measure. Membership inference attacks against predictive models have been studied extensively (Shokri et al., 2017; Baluta et al., 2022; Hu et al., 2022; Liu et al., 2022; He et al., 2022; Carlini et al., 2021a), and recent work has also developed membership inference attacks against synthetic data (Stadler et al., 2022; Chen et al., 2020).

In a reconstruction attack, the attacker is not presented with a real sample to classify as belonging to the training set or not, but rather has to *create* samples belonging to the training set based only on the algorithm's output. Reconstruction attacks have been successfully conducted against large language models (Carlini et al., 2021b). At present, these attacks require the attacker to have a great deal of auxiliary information to succeed. For our purposes, we are interested in privacy attacks to measure the privacy of an algorithm, and such a granular task may place too high burden on the attacker to accurately detect "small" amounts of privacy leakage.

In an attribute inference attack (Bun et al., 2021; Stadler et al., 2022), the attacker tries to infer a sensitive attribute from a particular sample, based on its non-sensitive attributes and the attacked algorithm output. It has been argued that attribute inference is really the entire goal of statistical learning, and therefore should not be considered a privacy violation (Bun et al., 2021; Jayaraman and Evans, 2022).

**Differential Privacy (DP)**    DP (Dwork et al., 2014) and its variants (Mironov, 2017; Dwork and Rothblum, 2016) offer strong, information-theoretic privacy guarantees. A DP (probabilistic) algorithm is one in which the probability law of its output does not change much if one sample in its

input is changed. That is, if $D$ and $D'$ are two *adjacent* datasets (i.e., two datasets which differ in exactly one element), then the algorithm $\mathcal{A}$ is $\varepsilon$-DP if $\mathbb{P}(\mathcal{A}(D) \in S) \leq e^\varepsilon \mathbb{P}(\mathcal{A}(D') \in S)$ for any subset $S$ of the output space. DP has many desirable properties, such as the ability to compose DP methods or post-process the output without losing guarantees. Many simple "wrapper" methods are also available for certifying DP. Among the simplest, the Laplace mechanism, adds Laplace noise to the algorithm output. The noise level must generally depend on the *sensitivity* of the base algorithm, which measures how much a single input sample can change the algorithm's output. The method we propose in this work is similar to the Laplace mechanism, but we show that the amount of noise needed can be reduced drastically. Abadi et al. (2016) introduced DP-SGD, a powerful tool enabling DP to be combined with deep learning, based on a small modification to the standard gradient descent training. However, enforcing DP does not come without a cost. Enforcing DP with high levels of privacy (small $\varepsilon$) often comes with sharp decreases in algorithm utility (Tao et al., 2021; Stadler et al., 2022). DP is also difficult to audit; it must be proven mathematically for a given algorithm. Checking it empirically is generally computationally intractable (Gilbert and McMillan, 2018). The difficulty of checking DP has led to widespread implementation issues (and even errors due to finite machine precision), which invalidate the guarantees of DP (Jagielski et al., 2020).

While the basic definition of DP can be difficult to interpret, equivalent "operational" definitions have been developed (Wasserman and Zhou, 2010; Kairouz et al., 2015; Nasr et al., 2021). These works show that DP can equivalently be expressed in terms of the maximum success rate on an adversary which seeks to distinguish between two adjacent datasets $D$ and $D'$, given only the output of a DP algorithm. While similar to the setting of membership inference at face value, there are subtle differences. In particular, in the case of membership inference, one must consider *all* datasets which could have contained the target record, and *all* datasets which do not contain the target record, and distinguish between the algorithm's output in this larger space of possibilities.

Lastly, the independent work of Thudi et al. (2022) specifically applies DP to bound membership inference rates, and our results in Sec. 3.4 complement theirs on the relationship between membership inference and DP. However, our results show that DP is not *required* to prevent membership inference; it is merely one option, and we give alternative methods for defending against membership inference.

## 3    Membership Inference Privacy

### 3.1    Notation

We make use of the following notation. We will use $\mathcal{D}$ to refer to our entire dataset, which consists of $n$ samples all of which must remain private. We will use $\mathbf{x} \in \mathcal{D}$ or $\mathbf{x}^* \in \mathcal{D}$ to refer to a particular sample. $\mathcal{D}_{\text{train}} \subseteq \mathcal{D}$ refers to a size-$k$ subset of $\mathcal{D}$. We will assume the subset is selected randomly, so $\mathcal{D}_{\text{train}}$ is a random variable. The remaining data $\mathcal{D} \setminus \mathcal{D}_{\text{train}}$ will be referred to as the holdout data. We denote by $\mathbb{D}$ the set of all size-$k$ subsets of $\mathcal{D}$ (i.e., all possible training sets), and we will use $D \in \mathbb{D}$ to refer to a particular realization of the random variable $\mathcal{D}_{\text{train}}$. Finally, given a particular sample $\mathbf{x}^* \in \mathcal{D}$, $\mathbb{D}^{\text{in}}$ (resp. $\mathbb{D}^{\text{out}}$) will refer to sets $D \in \mathbb{D}$ for which $\mathbf{x}^* \in D$ (resp. $\mathbf{x}^* \notin D$).

### 3.2    Theoretical Motivation

The implicit assumption behind the public release of any statistical algorithm–be it a generative or predictive ML model, or even the release of simple population statistics–is that it is acceptable for *statistical information about the modelled data* to be released publicly. In the context of membership inference, this poses a potential problem: if the population we are modeling is significantly different from the "larger" population, then if our algorithm's output contains any useful information whatsoever, it *should* be possible for an attacker to infer whether or not a given record could have plausibly come from our training data or not.

We illustrate this concept with an example. Suppose the goal is to publish an ML model which predicts a patient's blood pressure from several biomarkers, specifically for patients who suffer from a particular chronic disease. To do this, we collect a dataset of individuals with confirmed cases of the disease, and use this data to train a linear regression model with coefficients $\hat{\theta}$. Formally, we let $\mathbf{x} \in \mathbb{R}^d$ denote the features (e.g. biomarker values), $z \in \mathbb{R}$ denote the patient's blood pressure, and $y = \mathbb{1}\{\text{patient has the chronic disease in question}\}$. In this case, the private dataset $\mathcal{D}_{\text{train}}$ contains only the patients with $y = 1$. Assume that in the general populace, patient features are drawn from a

mixture model:

$$y \sim \text{Bernoulli}(p), \qquad \mathbf{x} \sim \mathcal{N}(0, I), \qquad z|\mathbf{x}, y \sim \theta_y^\top \mathbf{x}, \qquad \theta_0 \neq \theta_1.$$

In the membership inference attack scenario, an adversary observes a data point $(\mathbf{x}^*, z^*)$ and the model $\hat{\theta}$, and tries to determine whether or not $(\mathbf{x}^*, z^*) \in \mathcal{D}_{\text{train}}$. If $\theta_0$ and $\theta_1$ are well-separated, then an adversary can train an effective classifier to determine the corresponding label $\mathbb{1}\{(\mathbf{x}^*, z^*) \in \mathcal{D}_{\text{train}}\}$ for $(\mathbf{x}^*, z^*)$ by checking whether or not $z^* \approx \hat{\theta}^\top \mathbf{x}^*$. Since only data with $y = 1$ belong to $\mathcal{D}_{\text{train}}$, this provides a signal to the adversary as to whether or not $\mathbf{x}^*$ could have belonged to $\mathcal{D}_{\text{train}}$ or not. The point is that in this setting, this outcome is unavoidable if $\hat{\theta}$ is to provide any utility whatsoever. In other words:

*In order to preserve utility, membership inference privacy must be measured with respect to the distribution from which the private data are drawn.*

The example above motivates the following theoretical ideal for our membership inference setting. Let $\mathcal{D} = \{\mathbf{x}_i\}_{i=1}^n$ be the private dataset and suppose that $\mathbf{x}_i \overset{\text{i.i.d.}}{\sim} \mathcal{P}$ for some probability distribution $\mathcal{P}$. (Note: Here, $\mathbf{x}^*$ corresponds to the complete datapoint $(\mathbf{x}^*, z^*)$ in the example above.) Let $\mathcal{A}$ be our (randomized) algorithm, and denote its output by $\theta = \mathcal{A}(\mathcal{D})$. We generate a test point based on:

$$y^* \sim \text{Bernoulli}\left(1/2\right), \qquad \mathbf{x}^*|y^* \sim y^*\text{Unif}(\mathcal{D}_{\text{train}}) + (1 - y^*)\mathcal{P},$$

i.e. $\mathbf{x}^*$ is a fresh draw from $\mathcal{P}$ or a random element of the private training data with equal probability. Let $\mathcal{I}$ denote any membership inference algorithm which takes as input $\mathbf{x}^*$ and the algorithm's output $\theta = \mathcal{A}(\mathcal{D}_{\text{train}})$. The notion of privacy we wish to enforce is that $\mathcal{I}$ cannot do much better to ascertain the membership of $\mathbf{x}^*$ than guessing randomly:

$$\mathbb{P}_{\mathcal{A}, \mathcal{D}_{\text{train}}}(\mathcal{I}(\mathbf{x}^*, \theta) = y^*) \leq 1/2 + \eta, \tag{1}$$

where ideally $\eta \ll 1/2$.

### 3.3 Practical Definition

In reality, we do not have access to the underlying distribution $\mathcal{P}$. Instead, we propose to use a bootstrap sampling approach to approximate fresh draws from $\mathcal{P}$.

**Definition 1** (Membership Inference Privacy (MIP)). *Given fixed $k \leq n$, let $\mathcal{D}_{\text{train}} \subseteq \mathcal{D}$ be a size-$k$ subset chosen uniformly at random from the elements in $\mathcal{D}$. For $\mathbf{x}^* \in \mathcal{D}$, let $y^* = \mathbb{1}\{\mathbf{x}^* \in \mathcal{D}_{\text{train}}\}$. An algorithm $\mathcal{A}$ is $\eta$-MIP with respect to $\mathcal{D}$ if for any identification algorithm $\mathcal{I}$ and for every $\mathbf{x}^* \in \mathcal{D}$, we have*

$$\mathbb{P}(\mathcal{I}(\mathbf{x}^*, \mathcal{A}(\mathcal{D}_{\text{train}})) = y^*) \leq \max\left\{\frac{k}{n}, 1 - \frac{k}{n}\right\} + \eta.$$

*Here, the probability is taken over the uniformly random size-$k$ subset $\mathcal{D}_{\text{train}} \subseteq \mathcal{D}$, as well as any randomness in $\mathcal{A}$ and $\mathcal{I}$.*

Definition 1 states that given the output of $\mathcal{A}$, an adversary cannot determine whether a given point was in the holdout set or training set with probability more than $\eta$ better than always guessing the a priori more likely outcome. In the remainder of the paper, we will set $k = n/2$, so that $\mathcal{A}$ is $\eta$-MIP if an attacker cannot have average accuracy greater than $(1/2 + \eta)$. This gives the largest a priori entropy for the attacker's classification task, which creates the highest ceiling on how much of an advantage an attacker can possibly gain from the algorithm's output, and consequently the most accurate measurement of privacy leakage. The choice $k = n/2$ also keeps us as close as possible to the theoretical motivation in the previous subsection. We note that analogues of all of our results apply for general $k$.

The definition of MIP is phrased with respect to *any* classifier (whose randomness is independent of the randomness in $\mathcal{A}$; if the adversary knows the algorithm and the random seed, we are doomed). While this definition is compelling in that it shows a bound on what any attacker can hope to accomplish, the need to consider all possible attack algorithms makes it difficult to work with technically. The following proposition shows that MIP is equivalent to a simpler definition which does not need to simultaneously consider all identification algorithms $\mathcal{I}$.

**Proposition 2.** *Let $\mathbb{A} = \text{Range}(\mathcal{A})$ and let $\mu$ denote the probability law of $\mathcal{A}(\mathcal{D}_{\text{train}})$. Then $\mathcal{A}$ is $\eta$-MIP if and only if*

$$\int_{\mathbb{A}} \left( \max \left\{ \mathbb{P}(\mathbf{x}^* \in \mathcal{D}_{\text{train}} \mid \mathcal{A}(\mathcal{D}_{\text{train}}) = A), \mathbb{P}(\mathbf{x}^* \notin \mathcal{D}_{\text{train}} \mid \mathcal{A}(\mathcal{D}_{\text{train}}) = A) \right\} d\mu(A) \right) \leq \frac{1}{2} + \eta.$$

*Furthermore, the optimal adversary is given by*

$$\mathcal{I}(\mathbf{x}^*, A) = \mathbb{1}\{\mathbb{P}(\mathbf{x}^* \in \mathcal{D}_{\text{train}} \mid \mathcal{A}(\mathcal{D}_{\text{train}}) = A) \geq 1/2\}.$$

Proposition 2 makes precise the intuition that the optimal attacker should guess the more likely of $\mathbf{x}^* \in \mathcal{D}_{\text{train}}$ or $\mathbf{x}^* \notin \mathcal{D}_{\text{train}}$ conditional on the output of $\mathcal{A}$. The optimal attacker's overall accuracy is then computed by marginalizing this conditional statement.

Finally, MIP also satisfies a post-processing inequality similar to the classical result in DP (Dwork et al., 2014). This states that any local functions of a MIP algorithm's output cannot degrade the privacy guarantee.

**Theorem 3.** *Suppose that $\mathcal{A}$ is $\eta$-MIP, and let $f$ be any (potentially randomized, with randomness independent of $\mathcal{D}_{\text{train}}$) function. Then $f \circ \mathcal{A}$ is also $\eta$-MIP.*

*Proof.* Let $\mathcal{I}_f$ be any membership inference algorithm for $f \circ \mathcal{A}$. Define $\mathcal{I}_{\mathcal{A}}(\mathbf{x}^*, \mathcal{A}(\mathcal{D}_{\text{train}})) = \mathcal{I}_f(\mathbf{x}^*, f(\mathcal{A}(\mathcal{D}_{\text{train}})))$. Since $\mathcal{A}$ is $\eta$-MIP, we have

$$\frac{1}{2} + \eta \geq \mathbb{P}(\mathcal{I}_{\mathcal{A}}(\mathbf{x}^*, \mathcal{A}(\mathcal{D}_{\text{train}})) = y^*) = \mathbb{P}(\mathcal{I}_f(\mathbf{x}^*, f(\mathcal{A}(\mathcal{D}_{\text{train}}))) = y^*).$$

Thus, $f \circ \mathcal{A}$ is $\eta$-MIP by Definition 1. $\qquad \square$

For example, Theorem 3 is important for the application of MIP to generative model training – if we can guarantee that our generative model is $\eta$-MIP, then any output produced by it is $\eta$-MIP as well.

### 3.4 Relation to Differential Privacy

In this section, we make precise the relationship between MIP and the most common theoretical formulation of privacy: differential privacy (DP). We provide proof sketches for most of our results here; detailed proofs can be found in the Appendix. Our first theorem shows that DP is at least as strong as MIP.

**Theorem 4.** *Let $\mathcal{A}$ be $\varepsilon$-DP. Then $\mathcal{A}$ is $\eta$-MIP with $\eta = \frac{1}{1+e^{-\varepsilon}} - \frac{1}{2}$. Furthermore, this bound is tight, i.e. for any $\varepsilon > 0$, there exists an $\varepsilon$-DP algorithm against which the optimal attacker has accuracy $\frac{1}{1+e^{-\varepsilon}}$.*

To help interpret this result, we remark that for $\varepsilon \approx 0$, we have $\frac{1}{1+e^{-\varepsilon}} - \frac{1}{2} \approx \varepsilon/4$. Thus in the regime where strong privacy guarantees are required ($\eta \approx 0$), $\eta \approx \varepsilon/4$.

In fact, DP is *strictly* stronger than MIP, which we make precise with the following theorem.

**Theorem 5.** *For any $\eta > 0$, there exists an algorithm $\mathcal{A}$ which is $\eta$-MIP but not $\varepsilon$-DP for any $\varepsilon < \infty$.*

In order to better understand the difference between DP and MIP, let us again examine Proposition 2. Recall that this proposition showed that *marginally* over the output of $\mathcal{A}$, the conditional probability that $\mathbf{x}^* \in \mathcal{D}_{\text{train}}$ given the algorithm output should not differ too much from the unconditional probability that $\mathbf{x}^* \in \mathcal{D}_{\text{train}}$. The following proposition shows that DP requires this condition to hold for *every* output of $\mathcal{A}(\mathcal{D}_{\text{train}})$.

**Proposition 6.** *If $\mathcal{A}$ is an $\varepsilon$-DP algorithm, then for any $\mathbf{x}^*$, we have*

$$\frac{\mathbb{P}(\mathbf{x}^* \notin \mathcal{D}_{\text{train}} \mid \mathcal{A}(\mathcal{D}_{\text{train}}))}{\mathbb{P}(\mathbf{x}^* \in \mathcal{D}_{\text{train}} \mid \mathcal{A}(\mathcal{D}_{\text{train}}))} \leq e^{\varepsilon} \frac{\mathbb{P}(\mathbf{x}^* \notin \mathcal{D}_{\text{train}})}{\mathbb{P}(\mathbf{x}^* \in \mathcal{D}_{\text{train}})}.$$

Proposition 6 can be thought of as an extension of the Bayesian interpretation of DP explained by Jordon et al. (2022). Namely, the definition of DP immediately implies that, for any two adjacent sets $D$ and $D'$,

$$\frac{\mathbb{P}(\mathcal{D}_{\text{train}} = D \mid \mathcal{A}(\mathcal{D}_{\text{train}}))}{\mathbb{P}(\mathcal{D}_{\text{train}} = D' \mid \mathcal{A}(\mathcal{D}_{\text{train}}))} \leq e^{\varepsilon} \frac{\mathbb{P}(\mathcal{D}_{\text{train}} = D)}{\mathbb{P}(\mathcal{D}_{\text{train}} = D')}.$$

# 4 Guaranteeing MIP via Noise Addition

In this section, we show that a small modification to standard training procedures can be used to guarantee MIP. Suppose that $\mathcal{A}$ takes as input a data set $D$ and produces output $\theta \in \mathbb{R}^d$. For instance, $\mathcal{A}$ may compute a simple statistical query on $D$, such as mean estimation, but our results apply equally well in the case that e.g. $\mathcal{A}(D)$ are the weights of a neural network trained on $D$. If $\theta$ are the weights of a generative model, then if we can guarantee MIP for $\theta$, then by the data processing inequality (Theorem 3), this guarantees privacy for any output of the generative model.

The distribution over training data (in our case, the uniform distribution over size $n/2$ subsets of our complete dataset $\mathcal{D}$) induces a distribution over the output $\theta$. The question is, what is the smallest amount of noise we can add to $\theta$ which will guarantee MIP? If we add noise on the order of $\max_{D \sim D' \subseteq \mathcal{D}} \|\mathcal{A}(D) - \mathcal{A}(D')\|$, then we can adapt the standard proof for guaranteeing DP in terms of algorithm sensitivity to show that a restricted version of DP (only with respect subsets of $\mathcal{D}$) holds in this case, which in turn guarantees MIP. However, recall that by Propositions 2 and 6, MIP is only asking for a *marginal* guarantee on the change in the posterior probability of $D$ given $A$, whereas DP is asking for a *conditional* guarantee on the posterior. So while $\max$ seems necessary for a conditional guarantee, the *moments* of $\theta$ should be sufficient for a marginal guarantee. Theorem 7 shows that this intuition is correct.

**Theorem 7.** *Let $\|\cdot\|$ be any norm, and let $\sigma^M \geq \mathbb{E}\|\theta - \mathbb{E}\theta\|^M$ be an upper bound on the $M$-th central moment of $\theta$ with respect to this norm over the randomness in $\mathcal{D}_{\mathrm{train}}$ and $\mathcal{A}$. Let $X$ be a random variable with density proportional to $\exp(-\frac{1}{c\sigma}\|X\|)$ with $c = (7.5/\eta)^{1+\frac{2}{M}}$. Finally, let $\hat{\theta} = \theta + X$. Then $\hat{\theta}$ is $\eta$-MIP, i.e., for any adversary $\mathcal{I}$,*

$$\mathbb{P}(\mathcal{I}(\mathbf{x}^*, \hat{\theta}) = y^*) \leq 1/2 + \eta.$$

At first glance, Theorem 7 may appear to be adding noise of equal magnitude to all of the coordinates of $\theta$, regardless of how much each contributes to the central moment $\sigma$. However, by carefully selecting the norm $\|\cdot\|$, we can add non-isotropic noise to $\theta$ such that the marginal noise level reflects the variability of each specific coordinate of $\theta$. This is the content of Corollary 8. ($\mathrm{GenNormal}(\mu, \alpha, \beta)$ refers to the probability distribution with density proportional to $\exp(-((x - \mu)/\alpha)^\beta)$.)

**Corollary 8.** *Let $\sigma_i^M \geq \mathbb{E}|\theta_i - \mathbb{E}\theta_i|^M$, and define $\|x\|_{\sigma,M} = \left(\sum_{i=1}^d \frac{|x_i|^M}{d\sigma_i^M}\right)^{1/M}$. Generate*

$$Y_i \sim \mathrm{GenNormal}(0, \sigma_i, M), \quad U = Y/\|Y\|_{\sigma,M}$$

*and draw $r \sim \mathrm{Laplace}\left((6.16/\eta)^{1+2/M}\right)$. Finally, set $X = rU$ and return $\hat{\theta} = \theta + X$. Then $\hat{\theta}$ is $\eta$-MIP.*

In practice, the $\sigma_i$ may not be known, but they can easily be estimated from data. We implement this intuition and devise a practical method for guaranteeing MIP in Algorithm 1 in the appendix.

**When Does MIP Improve Over DP?** By Theorem 4, any DP algorithm gives rise to a MIP algorithm, so we *never* need to add more noise than the amount required to guarantee DP, in order to guarantee MIP. However, Theorem 7 shows that MIP affords an advantage over DP when the variance of our algorithm's output (over subsets of size $n/2$) is much smaller than its sensitivity $\Delta$, which is defined as the maximum change in the algorithm's output when evaluated on two datasets which differ in only one element. For instance, applying the Laplace mechanism from DP requires a noise which scales like $\Delta/\epsilon$ to guarantee $\varepsilon$-DP. It is easy to construct examples where the variance is much smaller than the sensitivity if the output of our "algorithm" is allowed to be completely arbitrary as a function of the input. However, it is more interesting to ask if there are any *natural* settings in which this occurs. Proposition 9 answers this question in the affirmative.

**Proposition 9.** *For any finite $D \subseteq \mathbb{R}$, define $\mathcal{A}(D) = \frac{1}{\sum_{x \in D} x}$. Given a dataset $\mathcal{D}$ of size $n$, define $\mathbb{D} = \{D \subseteq \mathcal{D} : |D| = \lfloor n/2 \rfloor\}$, and define*

$$\sigma^2 = \mathrm{Var}(\mathcal{A}(D)), \qquad \Delta = \max_{D \sim D' \in \mathbb{D}} |\mathcal{A}(D) - \mathcal{A}(D')|.$$

*Here the variance is taken over $D \sim \mathrm{Unif}(\mathbb{D})$. Then for all $n$, there exists a dataset $|\mathcal{D}| = n$ such that $\sigma^2 = O(1)$ but $\Delta = \Omega(2^{n/3})$.*

# 5 Simulation Results

## 5.1 Noise Level in Proposition 9

To illustrate our theoretical results, we plot the noise level needed to guarantee MIP vs. the corresponding level of DP (with the correspondence given by Theorem 4) for the example in Proposition 9.

Refer to Fig. 1. Dotted lines refer to DP, while the solid line is for MIP with $M = 2$. The $x$-axis gives the best possible bound on the attacker's improvement in accuracy over random guessing–i.e., the parameter $\eta$ for an $\eta$-MIP method–according to that method's guarantees. The $y$-axis denotes the amount of noise that must be added to the non-private algorithm's output, as measured by the scale parameter of the Laplace noise that must be added (lower is better). For DP, the amount of noise necessary changes with the size $n$ of the private dataset. For MIP, the amount of noise does not change, so there is only one line. The results show that for even small datasets ($n \geq 36$) and for $\eta \geq 0.01$, direct noise accounting for MIP gives a large advantage over guaranteeing MIP via DP. In practice, such small datasets are uncommon. As $n$ increases above even this modest range, the advantage in terms of noise reduction for MIP vs. DP quickly becomes many orders of magnitude and is not visible on the plot. (Refer to Proposition 9. The noise required for DP grows exponentially in $n$, while it remains constant in $n$ for MIP.)

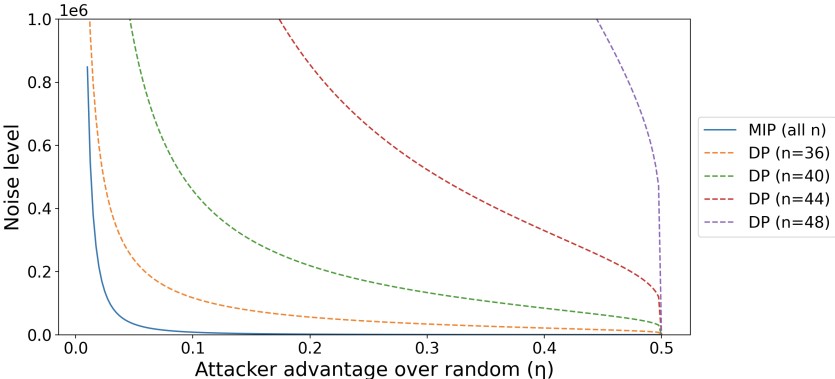

Figure 1: Noise level vs. privacy guarantee for MIP and DP (lower is better). For datasets with at least $n = 36$ points and for almost all values of $\eta$, MIP allows us to add much less noise than what would be required by naively applying DP.

## 5.2 Synthetic Data Generation

We conduct an experiment as a representation scenario for private synthetic data generation. We are given a dataset which consists of i.i.d. draws from a private ground truth distribution $P$. Our goal is to learn a generative model $G$ which allows us to (approximately) sample from $P$. That is, given a latent random variable $z$ (we will take $z \sim \mathcal{N}(0, I)$), we have $G(z) \sim P$. If $G$ is itself private and a good approximation for $P$, then by the post-processing theorem (Theorem 3), we can use $G$ to generate synthetic draws from $P$, without violating the privacy of any sample in the training data. For this experiment, we will take $G$ to be a single-layer linear network with no bias, i.e. $G(z) = Wz$ for some weight matrix $W$. The ground truth distribution $P = \mathcal{N}(0, \Sigma)$ is a mean-0 normal distribution with (unknown) non-identity covariance $\Sigma$. In this setting, $W = \Sigma^{1/2}$ would exactly reproduce $P$.

Let $\{\mathbf{x}_i\}_{i=1}^n \subseteq \mathbb{R}^d$ be the training data. Rather than attempting to learn $W \approx \Sigma^{1/2}$ directly, we will instead try to learn $A \approx \Sigma$. We can then set $W = (A + A^\top)^{1/2}$. If $A \approx \Sigma$, then we will have $W \approx \Sigma^{1/2}$ and $G(z) \approx P$. We learn $A$ by minimizing the objective $\min_A \left\| A - \frac{1}{n} \sum_{i=1}^n \mathbf{x}_i \mathbf{x}_i^\top \right\|_F^2$ via gradient descent. For the DP method, we implement DP-SGD (Abadi et al., 2016) with a full batch. We chose the clipping parameter $C$ according to the authors' recommendations, i.e. the median of the unclipped gradients over the training procedure. To implement MIP, we used Corollary 8 with $M \in \{2, 4, 6\}$ and the corresponding $\sigma_i$'s computed empirically over 128 random train/holdout splits of the base dataset. The results below use $d = 3$ and $n = 500,000$.

Refer to Figure 2. The $x$-axes show the theoretical privacy level $\eta$ (again using the tight correspondence between $\varepsilon$ and $\eta$ from Theorem 4), and the $y$-axes show the relative error $\|A - \Sigma\|_F / \|\Sigma\|_F$ (lower is better). The three plots show the same results zoomed in on different ranges for $\eta$ to see greater granularity, and the shaded region shows the standard error of the mean over 10 runs. For $\eta \geq 0.1$, MIP with any of the tested values of $M$ outperforms DP in terms of accuracy. For the entire tested $\eta$ range (the smallest of which was $\eta = 0.01$), MIP with $M = 4$ or 6 outperforms DP. (The first plot does not show MIP with $M = 2$ because its error is very large when $\eta < 0.1$.) Finally, observe that DP is *never* able to obtain relative error less than 1 (meaning the resulting output is mostly noise), while MIP obtains relative error less than 1 for $\eta \approx 0.2$ and larger.

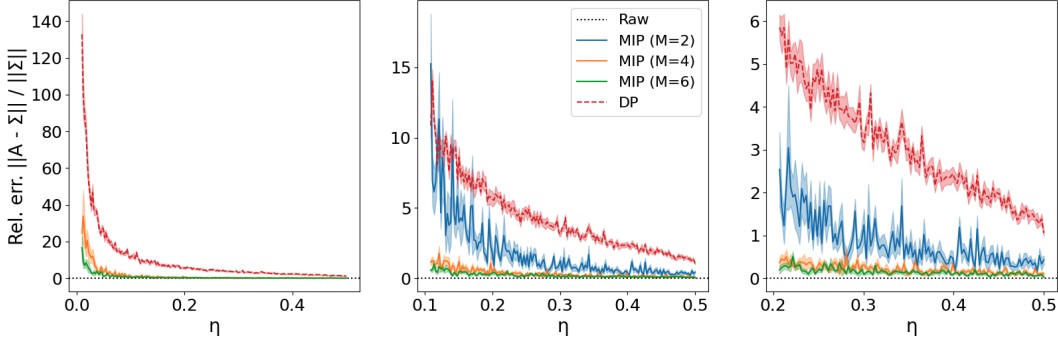

Figure 2: Error vs. privacy guarantee for MIP and DP (lower is better). Raw refers to the error of the non-private base algorithm, which just computes $A$ by vanilla gradient descent. MIP improves over DP in terms of accuracy when $\eta \geq 0.1$ for all of the tested values of $M \in \{2, 4, 6\}$, and for $M \in \{4, 6\}$, MIP improves of DP for the entire range of $\eta$. Note that MIP obtains relative error $< 1$ for some $\eta$, while DP always has relative error larger than 1. MIP with $M = 2$ is not shown in the first plot because the error is large in the small $\eta$ range, obscuring the results for the other methods.

## 6   Conclusion

In this work, we proposed a novel privacy property, membership inference privacy (MIP) and explained its properties and relationship with differential privacy (DP). The MIP property is more readily interpretable than the guarantees offered by (DP). MIP also requires a smaller amount of noise to guarantee as compared to DP, and therefore can retain greater utility in practice. We proposed a simple "wrapper" method for guaranteeing MIP, which can be implemented with a minor modification both to simple statistical queries or more complicated tasks such as the training procedure for parametric machine learning models.

**Limitations**   As the example used to prove Theorem 5 shows, there are cases where apparently non-private algorithms can satisfy MIP. Thus, algorithms which satisfy MIP may require post-processing to ensure that the output is not one of the low-probability events in which data privacy is leaked. In addition, because MIP is determined with respect to a holdout set still drawn from $\mathcal{D}$, an adversary may be able to determine with high probability whether or not a given sample was contained in $\mathcal{D}$, rather than just in $\mathcal{D}_{\mathrm{train}}$, if $\mathcal{D}$ is sufficiently different from the rest of the population.

**Future Work**   Theorem 4 suggests that DP implies MIP in general. However, Theorem 7 shows that a finer-grained analysis of a standard DP mechanism (the Laplace mechanism) is possible, showing that we can guarantee MIP with less noise. It seems plausible that a similar analysis can be undertaken for other DP mechanisms. In addition to these "wrapper" type methods which can be applied on top of existing algorithms, bespoke algorithms for guaranteeing MIP in particular applications (such as synthetic data generation) are also of interest. Noise addition is a simple and effective way to enforce privacy, but other classes of mechanisms may also be possible. For instance, is it possible to directly regularize a probabilistic model using Proposition 2? Finally, the connections between MIP and other theoretical notions of privacy (Renyi DP (Mironov, 2017), concentrated DP (Dwork and Rothblum, 2016), etc.) are also of interest. Lastly, this paper focused on developing on the theoretical principles and guarantees of MIP, but systematic empirical evaluation is an important direction for future work.

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

# A   Practical Algorithm for Guaranteeing MIP

We remark briefly that the estimator for $\sigma_j$ used in Algorithm 1 is not unbiased, but it is consistent (i.e., the bias approaches 0 as $B \to \infty$). When $M = 2$, there is a well-known unbiased estimate for the variance which replace $1/B$ with $1/(B-1)$, and one can make similar corrections for general $M$ (Gerlovina and Hubbard, 2019). In practice, these corrections yield very small difference and the naive estimator presented in the algorithm should suffice.

---

**Algorithm 1** MIP via noise addition

---

**Require:** Private dataset $\mathcal{D}$, $\sigma$ estimation budget $B$, MIP parameter $\eta$
  $\mathcal{D}_{\text{train}} \leftarrow \text{RANDOMSPLIT}(\mathcal{D}, 1/2)$

  *# Estimate $\sigma$ if an a priori bound is not known*
  **for** $i = 1, \ldots, B$ **do**
    $\mathcal{D}_{\text{train}}^{(i)} \leftarrow \text{RANDOMSPLIT}(\mathcal{D}_{\text{train}}, 1/2)$
    $\theta^{(i)} \leftarrow \mathcal{A}(\mathcal{D}_{\text{train}}^{(i)})$
  **end for**
  **for** $j = 1, \ldots, d$ **do**
    $\bar{\theta}_j \leftarrow \frac{1}{B} \sum_{i=1}^{B} \theta_j^{(i)}$
    $\sigma_j \leftarrow \left( \frac{1}{B} \sum_{i=1}^{B} (\theta_j^{(i)} - \bar{\theta}_j)^M \right)^{1/M}$
  **end for**

  *# Add appropriate noise to the base algorithm's output*
  $U \leftarrow \text{Unif}(\{u \in \mathbb{R}^d \; : \; \|u\|_{\sigma,M} = 1\})$
  $r \leftarrow \text{Laplace}\left( \left( \frac{6.16}{\eta} \right)^{1+2/M} \right)$
  $X \leftarrow rU$
  **return** $\mathcal{A}(\mathcal{D}_{\text{train}}) + X$

---

# B   Deferred Proofs

For the reader's convenience, we restate all lemmas, theorems, etc. here.

**Proposition 2.** *Let $\mathbb{A} = \text{Range}(\mathcal{A})$ and let $\mu$ denote the probability law of $\mathcal{A}(\mathcal{D}_{\text{train}})$. Then $\mathcal{A}$ is $\eta$-MIP if and only if*

$$\int_{\mathbb{A}} \left( \max \left\{ \mathbb{P}(\mathbf{x}^* \in \mathcal{D}_{\text{train}} \mid \mathcal{A}(\mathcal{D}_{\text{train}}) = A), \mathbb{P}(\mathbf{x}^* \notin \mathcal{D}_{\text{train}} \mid \mathcal{A}(\mathcal{D}_{\text{train}}) = A) \right\} d\mu(A) \right) \leq \frac{1}{2} + \eta.$$

*Furthermore, the optimal adversary is given by*

$$\mathcal{I}(\mathbf{x}^*, A) = \mathbb{1}\{\mathbb{P}(\mathbf{x}^* \in \mathcal{D}_{\text{train}} \mid \mathcal{A}(\mathcal{D}_{\text{train}}) = A) \geq 1/2\}.$$

*Proof.* We will show that the membership inference algorithm $\mathcal{I}(\mathbf{x}^*, A) = \mathbb{1}\{\mathbb{P}(\mathbf{x}^* \in \mathcal{D}_{\text{train}} \mid \mathcal{A}(\mathcal{D}_{\text{train}}) = A) \geq 1/2\}$ is optimal, then compute the resulting probability of member-

ship inference. We have

$$\mathbb{P}(\mathcal{I}(\mathbf{x}^*, \mathcal{A}(\mathcal{D}_{\text{train}})) = y^*) = \sum_{\mathcal{D}_{\text{train}} \subseteq \mathcal{D}} \binom{n}{k}^{-1} \sum_{A \in \mathbb{A}} \mathbb{P}(\mathcal{A}(\mathcal{D}_{\text{train}}) = A) \cdot \mathbb{P}(\mathcal{I}(\mathbf{x}^*, A) = \mathbb{1}\{\mathbf{x}^* \in \mathcal{D}_{\text{train}}\})$$

$$= \binom{n}{k}^{-1} \sum_{A \in \mathbb{A}} \left[ \sum_{D \in \mathbb{D}^{\text{in}}} \mathbb{P}(\mathcal{A}(D) = A) \cdot \mathbb{P}(\mathcal{I}(\mathbf{x}^*, A) = 1) \right.$$

$$\left. + \sum_{D \in \mathbb{D}^{\text{out}}} \mathbb{P}(\mathcal{A}(D) = A) \cdot (1 - \mathbb{P}(\mathcal{I}(\mathbf{x}^*, A) = 1)) \right]$$

$$= \binom{n}{k}^{-1} \sum_{A \in \mathbb{A}} \left[ \left( \sum_{D \in \mathbb{D}^{\text{in}}} \mathbb{P}(\mathcal{A}(D) = A) - \sum_{D \in \mathbb{D}^{\text{out}}} \mathbb{P}(\mathcal{A}(D) = A) \right) \mathbb{P}(\mathcal{I}(\mathbf{x}^*, A) = 1) \right.$$

$$\left. + \sum_{D \in \mathbb{D}^{\text{out}}} \mathbb{P}(\mathcal{A}(D) = A) \right].$$

The choice of algorithm $\mathcal{I}$ just specifies the value of $\mathbb{P}(\mathcal{I}(\mathbf{x}^*, A) = 1)$ for each sample $\mathbf{x}^*$ and each $A \in \mathbb{A}$. We see that the maximum membership inference probability is obtained when

$$\mathbb{P}(\mathcal{I}(\mathbf{x}^*, A) = 1) = \mathbb{1} \left\{ \sum_{D \in \mathbb{D}^{\text{in}}} \mathbb{P}(\mathcal{A}(D) = A) - \sum_{D \in \mathbb{D}^{\text{out}}} \mathbb{P}(\mathcal{A}(D) = A) \geq 0 \right\}, \qquad (2)$$

which implies that

$$\sum_{D \in \mathbb{D}^{\text{in}}} \mathbb{P}(\mathcal{A}(D) = A) - \sum_{D \in \mathbb{D}^{\text{out}}} \mathbb{P}(\mathcal{A}(D) = A) \leq \binom{n}{k}^{-1} \sum_{A \in \mathbb{A}} \max \left\{ \sum_{D \in \mathbb{D}^{\text{in}}} \mathbb{P}(\mathcal{A}(D) = A), \sum_{D \in \mathbb{D}^{\text{out}}} \mathbb{P}(\mathcal{A}(D) = A) \right\}.$$
$$(3)$$

To conclude, observe that

$$\mathbb{P}(\mathbf{x}^* \in D \mid \mathcal{A}(D) = A) = \frac{\mathbb{P}(x \in D \wedge \mathcal{A}(D) = A)}{\mathbb{P}(\mathcal{A}(D) = A)} = \frac{\sum_{D \in \mathbb{D}^{\text{in}}} \binom{n}{k}^{-1} \mathbb{P}_{\mathcal{A}}(\mathcal{A}(D) = A)}{\mathbb{P}_{\mathcal{A},D}(\mathcal{A}(D) = A)}. \qquad (4)$$

The result follows by rearranging the expression (4) and plugging it into (2) and (3). □

In what follows, we will assume without loss of generality that $k \geq n/2$. The proofs in the case $k < n/2$ are almost identical and can be obtained by simply swapping $\mathbb{D}^{\text{in}} \leftrightarrow \mathbb{D}^{\text{out}}$ and $k \leftrightarrow n - k$.

**Lemma 10.** *Fix $\mathbf{x}^* \in \mathcal{D}$ and let $\mathbb{D}^{\text{in}} = \{D \in \mathcal{D} \mid \mathbf{x}^* \in D\}$ and $\mathbb{D}^{\text{out}} = \{D \in \mathcal{D} \mid \mathbf{x}^* \notin D\}$. If $k \geq n/2$ then there is an injective function $f : \mathbb{D}^{\text{out}} \to \mathbb{D}^{\text{in}}$ such that $D \sim f(D)$ for all $D \in \mathbb{D}^{\text{out}}$.*

*Proof.* We define a bipartite graph $G$ on nodes $\mathbb{D}^{\text{in}}$ and $\mathbb{D}^{\text{out}}$. There is an edge between $D^{\text{in}} \in \mathbb{D}^{\text{in}}$ and $D^{\text{out}} \in \mathbb{D}^{\text{out}}$ if $D^{\text{out}}$ can be obtained from $D^{\text{in}}$ by removing $\mathbf{x}^*$ from $D^{\text{in}}$ and replacing it with another element, i.e. if $D^{\text{in}} \sim D^{\text{out}}$. To prove the lemma, it suffices to show that there is a matching on $G$ which covers $D^{\text{out}}$. We will show this via Hall's marriage theorem.

First, observe that $G$ is a $(k, n - k)$-biregular graph. Each $D^{\text{in}} \in \mathbb{D}^{\text{in}}$ has $n - k$ neighbors which are obtained from $D^{\text{in}}$ by selecting which of the remaining $n - k$ elements to replace $\mathbf{x}^*$ with; each $D^{\text{out}} \in \mathbb{D}^{\text{out}}$ has $k$ neighbors which are obtained by selecting which of the $k$ elements in $D^{\text{out}}$ to replace with $\mathbf{x}^*$.

Let $W \subseteq \mathbb{D}^{\mathrm{out}}$ and let $N(W) \subseteq \mathbb{D}^{\mathrm{in}}$ denote the neighborhood of $W$. We have the following:

$$|N(W)| = \sum_{D^{\mathrm{in}} \in N(W)} \frac{\sum_{D^{\mathrm{out}} \in W} \mathbb{1}\{D^{\mathrm{out}} \sim D^{\mathrm{in}}\}}{\sum_{D^{\mathrm{out}} \in W} \mathbb{1}\{D^{\mathrm{out}} \sim D^{\mathrm{in}}\}}$$

$$\geq \sum_{D^{\mathrm{in}} \in N(W)} \frac{\sum_{D^{\mathrm{out}} \in W} \mathbb{1}\{D^{\mathrm{out}} \sim D^{\mathrm{in}}\}}{\sum_{D^{\mathrm{out}} \in \mathbb{D}^{\mathrm{out}}} \mathbb{1}\{D^{\mathrm{out}} \sim D^{\mathrm{in}}\}}$$

$$= \frac{1}{n-k} \sum_{D^{\mathrm{out}} \in W} \sum_{D^{\mathrm{in}} \in N(W)} \mathbb{1}\{D^{\mathrm{out}} \sim D^{\mathrm{in}}\} \tag{5}$$

$$= \frac{k}{n-k}|W|. \tag{6}$$

Equation (5) holds since each $D^{\mathrm{in}}$ has degree $n-k$ and by exchanging the order of summation. Similarly, (6) holds since each $D^{\mathrm{out}}$ has degree $k$. When $k \geq n/2$, we thus have $|N(W)| \geq |W|$ for every $W \subseteq \mathbb{D}^{\mathrm{out}}$ and the result follows by Hall's marriage theorem. $\qquad \square$

**Theorem 4.** *Let $\mathcal{A}$ be $\varepsilon$-DP. Then $\mathcal{A}$ is $\eta$-MIP with $\eta = \frac{1}{1+e^{-\varepsilon}} - \frac{1}{2}$. Furthermore, this bound is tight, i.e. for any $\varepsilon > 0$, there exists an $\varepsilon$-DP algorithm against which the optimal attacker has accuracy $\frac{1}{1+e^{-\varepsilon}}$.*

*Proof.* Let $f : \mathbb{D}^{\mathrm{out}} \to \mathbb{D}^{\mathrm{in}}$ denote the injection guaranteed by Lemma 10. We have

$$\mathbb{P}(\mathcal{I}(\mathbf{x}^*, \mathcal{A}(D)) = y^*) = \frac{1}{\binom{n}{k}} \left[ \sum_{D^{\mathrm{in}} \in \mathbb{D}^{\mathrm{in}}} \mathbb{P}(\mathcal{I}(\mathbf{x}^*, \mathcal{A}(D^{\mathrm{in}})) = 1) + \sum_{D^{\mathrm{out}} \in \mathbb{D}^{\mathrm{out}}} \mathbb{P}(\mathcal{I}(\mathbf{x}^*, \mathcal{A}(D^{\mathrm{out}})) = 0) \right]$$

$$\leq \frac{1}{\binom{n}{k}} \left[ \sum_{D^{\mathrm{in}} \in \mathbb{D}^{\mathrm{in}}} \mathbb{P}(\mathcal{I}(\mathbf{x}^*, \mathcal{A}(D^{\mathrm{in}})) = 1) + \sum_{D^{\mathrm{out}} \in \mathbb{D}^{\mathrm{out}}} \left( e^{\varepsilon} \mathbb{P}(\mathcal{I}(\mathbf{x}^*, \mathcal{A}(f(D^{\mathrm{out}}))) = 0) + \delta \right) \right]$$

$$\leq \frac{1}{\binom{n}{k}} \left[ \sum_{D^{\mathrm{in}} \in \mathbb{D}^{\mathrm{in}}} \left( \mathbb{P}(\mathcal{I}(\mathbf{x}^*, \mathcal{A}(D^{\mathrm{in}})) = 1) + e^{\varepsilon} \mathbb{P}(\mathcal{I}(\mathbf{x}^*, \mathcal{A}(D^{\mathrm{in}})) = 0) \right) + \delta \binom{n-1}{k} \right] \tag{7}$$

$$\leq \frac{1}{\binom{n}{k}} \left[ e^{\varepsilon} \sum_{D^{\mathrm{in}} \in \mathbb{D}^{\mathrm{in}}} \left( \mathbb{P}(\mathcal{I}(\mathbf{x}^*, \mathcal{A}(D^{\mathrm{in}})) = 1) + \mathbb{P}(\mathcal{I}(\mathbf{x}^*, \mathcal{A}(D^{\mathrm{in}})) = 0) \right) + \delta \binom{n-1}{k} \right]$$

$$= \frac{1}{\binom{n}{k}} \left[ e^{\varepsilon} \binom{n-1}{k-1} + \delta \binom{n-1}{k} \right] = e^{\varepsilon} \frac{k}{n} + \delta \frac{n-k}{n}.$$

Here inequality (7) critically uses the fact that $f$ is injective, so at most one term from the sum over $\mathbb{D}^{\mathrm{out}}$ is added to each term in the sum over $\mathbb{D}^{\mathrm{in}}$. This completes the proof. $\qquad \square$

**Theorem 5.** *For any $\eta > 0$, there exists an algorithm $\mathcal{A}$ which is $\eta$-MIP but not $\varepsilon$-DP for any $\varepsilon < \infty$.*

*Proof.* Let $\mathcal{A}$ be defined as follows. Given a training set $D$, $\mathcal{A}(D)$ outputs a random subset of $D$ where each element is included independently and with probability $p$. It is obvious that such an algorithm is not $(\varepsilon, 0)$-DP for any $\varepsilon < \infty$: if $\mathbf{x} \in D$, then $\mathcal{A}(D) \in \{\{\mathbf{x}\}\}$ with positive probability. But if we replace $\mathbf{x}$ with $\mathbf{x}' \neq \mathbf{x}$ and call this adjacent dataset $D'$ (so that $\mathbf{x} \notin D'$, then $\mathcal{A}(D') \in \{\{\mathbf{x}\}\}$ with probability 0. Thus $\mathcal{A}$ is not differentially private for any $p > 0$.

We now claim that $\mathcal{A}$ is $(\varepsilon, 0)$-MIP for any $\varepsilon > 0$, provided that $p$ is small enough. To see this, observe the following. For any identification algorithm $\mathcal{I}$,

$$\mathbb{P}(\mathcal{I}(\mathbf{x}^*, \mathcal{A}(D)) = y^*) = \sum_{\mathcal{A}(D)} \left[ \mathbb{P}(\mathbf{x}^* \in D) \cdot \mathbb{P}(\mathcal{A}(D) \mid \mathbf{x}^* \in D) \cdot \mathbb{P}(\mathcal{I}(\mathbf{x}^*, \mathcal{A}(D)) = 1) \right.$$

$$\left. + \mathbb{P}(\mathbf{x}^* \notin D) \cdot \mathbb{P}(\mathcal{A}(D) \mid \mathbf{x}^* \notin D) \cdot (1 - \mathbb{P}(\mathcal{I}(\mathbf{x}^*, \mathcal{A}(D)) = 1)) \right]$$

$$= \sum_{\mathcal{A}(D)} \left[ \left( \frac{k}{n} \mathbb{P}(\mathcal{A}(D) \mid \mathbf{x}^* \in D) + (1 - \frac{k}{n}) \mathbb{P}(\mathcal{A}(D) \mid \mathbf{x}^* \notin D) \right) \mathbb{P}(\mathcal{I}(\mathbf{x}^*, \mathcal{A}(D)) = 1) \right.$$

$$\left. + (1 - \frac{k}{n}) \mathbb{P}(\mathcal{A}(D) \mid \mathbf{x}^* \notin D) \right]$$

$$\leq \sum_{\mathcal{A}(D)} \max \left\{ \frac{k}{n} \mathbb{P}(\mathcal{A}(D) \mid \mathbf{x}^* \in D), (1 - \frac{k}{n}) \mathbb{P}(\mathcal{A}(D) \mid \mathbf{x}^* \notin D) \right\}$$

$$\leq (1-p)^k \max \left\{ \frac{k}{n}, 1 - \frac{k}{n} \right\} + \sum_{\mathcal{A}(D) \neq \emptyset} (1 - (1-p)^k) \max \left\{ \frac{k}{n}, 1 - \frac{k}{n} \right\} \tag{8}$$

$$= \max \left\{ \frac{k}{n}, 1 - \frac{k}{n} \right\} \left[ (1-p)^k + C_{n,k}(1 - (1-p)^k) \right]. \tag{9}$$

Inequality (8) holds because $\mathcal{A}(D) = \emptyset$ with probability $(1-p)^k$ regardless of whether of not $\mathbf{x}^* \in D$, and therefore the probability that $\mathcal{A}(D) \neq \emptyset$ is at most $1 - (1-p)^k$ (again regardless of $\mathbf{x}^* \in D$ or not) for any $\mathcal{A}(D) \neq \emptyset$. The constant $C_{n,k}$ simply counts the number of possible $\mathcal{A}(D) \neq \emptyset$, which depends only on $n$ and $k$ but not $p$. Thus as $p \to 0$, (9) $\to 1$. This completes the proof. $\qquad \square$

The proof of Theorem 5 emphasizes that the membership inference privacy guarantee is *marginal* over the ouput of $\mathcal{A}$. Conditional on a particular output, an adversary may be able to determine whether or not $\mathbf{x}^* \in D$ with arbitrarily high precision. This is in contrast with the result of Proposition 6, which shows that even *conditionally* on a particular output of a DP algorithm, the adversary cannot gain too much.

**Proposition 6.** *If $\mathcal{A}$ is an $\varepsilon$-DP algorithm, then for any $\mathbf{x}^*$, we have*

$$\frac{\mathbb{P}(\mathbf{x}^* \notin \mathcal{D}_{\mathrm{train}} \mid \mathcal{A}(\mathcal{D}_{\mathrm{train}}))}{\mathbb{P}(\mathbf{x}^* \in \mathcal{D}_{\mathrm{train}} \mid \mathcal{A}(\mathcal{D}_{\mathrm{train}}))} \leq e^\varepsilon \frac{\mathbb{P}(\mathbf{x}^* \notin \mathcal{D}_{\mathrm{train}})}{\mathbb{P}(\mathbf{x}^* \in \mathcal{D}_{\mathrm{train}})}.$$

*Proof.* Using expression (4) (and the corresponding expression for $\mathbf{x}^* \notin \mathcal{D}_{\mathrm{train}}$), we have

$$\frac{\mathbb{P}(\mathbf{x}^* \notin \mathcal{D}_{\mathrm{train}} \mid \mathcal{A}(\mathcal{D}_{\mathrm{train}}) = A)}{\mathbb{P}(\mathbf{x}^* \in \mathcal{D}_{\mathrm{train}} \mid \mathcal{A}(\mathcal{D}_{\mathrm{train}}))} = \frac{\sum_{D \in \mathbb{D}^{\mathrm{out}}} \mathbb{P}_\mathcal{A}(\mathcal{A}(D) = A)}{\sum_{D \in \mathbb{D}^{\mathrm{in}}} \mathbb{P}_\mathcal{A}(\mathcal{A}(D) = A)}$$

$$\leq \frac{e^\varepsilon \sum_{D \in \mathbb{D}^{\mathrm{out}}} \min_{D' \in \mathbb{D}^{\mathrm{in}}, D' \sim D} \mathbb{P}_\mathcal{A}(\mathcal{A}(D) = A)}{\sum_{D \in \mathbb{D}^{\mathrm{in}}} \mathbb{P}_\mathcal{A}(\mathcal{A}(D) = A)}.$$

We now analyze this latter expression. We refer again to the biregular graph $G$ defined in Lemma 10. For $D \in \mathbb{D}^{\mathrm{out}}$, $N(D) \subseteq \mathbb{D}^{\mathrm{in}}$ refers to the neighbors of $D$ in $G$, and recall that $|N(D)| = k$ for all $D \in \mathbb{D}^{\mathrm{out}}$. Note that since each $D' \in \mathbb{D}^{\mathrm{in}}$ has $n - k$ neighbors, we have

$$\sum_{D \in \mathbb{D}^{\mathrm{out}}} \sum_{D' \in N(D)} \mathbb{P}(\mathcal{A}(D') = A) = (n - k) \sum_{D' \in \mathbb{D}^{\mathrm{in}}} \mathbb{P}(\mathcal{A}(D') = A).$$

Using this equality, we have

$$\frac{\sum_{D\in\mathbb{D}^{\text{out}}}\min_{D'\in\mathbb{D}^{\text{in}},D'\sim D}\mathbb{P}_{\mathcal{A}}(\mathcal{A}(D)=A)}{\sum_{D\in\mathbb{D}^{\text{in}}}\mathbb{P}_{\mathcal{A}}(\mathcal{A}(D)=A)}=\frac{\sum_{D\in\mathbb{D}^{\text{out}}}\min_{D'\in N(D)}\mathbb{P}_{\mathcal{A}}(\mathcal{A}(D)=A)}{\frac{1}{n-k}\sum_{D\in\mathbb{D}^{\text{out}}}\underbrace{\sum_{D'\in N(D)}\mathbb{P}_{\mathcal{A}}(\mathcal{A}(D)=A)}_{\geq k\min_{D'\in N(D)}P_{\mathcal{A}}(\mathcal{A}(D)=A)}}$$

$$\leq\frac{n-k}{k}.$$

Since $\mathbb{P}(\mathbf{x}^*\notin\mathcal{D}_{\text{train}})=\binom{n-1}{k}/\binom{n}{k}$ and $\mathbb{P}(\mathbf{x}^*\in\mathcal{D}_{\text{train}})=\binom{n-1}{k-1}/\binom{n}{k}$, we have $\frac{\mathbb{P}(\mathbf{x}^*\notin\mathcal{D}_{\text{train}})}{\mathbb{P}(\mathbf{x}^*\in\mathcal{D}_{\text{train}})}=\frac{n-k}{k}$. This completes the proof. $\qquad\square$

We remark that the proof of Proposition 6 indicates that converting between the case of distinguishing between two adjacent datasets (as in the inequality above, and as done in (Wasserman and Zhou, 2010; Kairouz et al., 2015; Nasr et al., 2021)) vs. the case of membership inference is non-trivial: both our proof and a similar one by Thudi et al. (2022) require the construction of a injective function between sets which do/do not contain $\mathbf{x}^*$.

**Theorem 7.** *Let $\|\cdot\|$ be any norm, and let $\sigma^M\geq\mathbb{E}\|\theta-\mathbb{E}\theta\|^M$ be an upper bound on the $M$-th central moment of $\theta$ with respect to this norm over the randomness in $\mathcal{D}_{\text{train}}$ and $\mathcal{A}$. Let $X$ be a random variable with density proportional to $\exp(-\frac{1}{c\sigma}\|X\|)$ with $c=(7.5/\eta)^{1+\frac{2}{M}}$. Finally, let $\hat{\theta}=\theta+X$. Then $\hat{\theta}$ is $\eta$-MIP, i.e., for any adversary $\mathcal{I}$,*

$$\mathbb{P}(\mathcal{I}(\mathbf{x}^*,\hat{\theta})=y^*)\leq 1/2+\eta.$$

*Proof.* We will assume that $k=n/2$ is an integer. Let $N=|\mathbb{D}^{\text{in}}|=|\mathbb{D}^{\text{out}}|$, and let $\mathbb{D}^{\text{in}}=\{D_1,\ldots,D_N\}$ and $\mathbb{D}^{\text{out}}=\{D'_1,\ldots,D'_N\}$. Define $a_i=\mathcal{A}(D_i)$ for $D_i\in\mathbb{D}^{\text{in}}$ and $b_j=\mathcal{A}(D'_j)$ for $D'_j\in\mathbb{D}^{\text{out}}$. Let $Z$ be a random variable which is uniformly distributed on $\{a_i\}\cup\{b_j\}$. We may assume without loss of generality that $\mathbb{E}Z=0$. In what follows, $c,\alpha,\beta$, and $\gamma$ are constants which we will choose later to optimize our bounds. We also make repeated use of the inequalities $1+x\leq e^x$ for all $x$; $\frac{1}{1+x}\geq 1-x$ for all $x\geq 0$; and $e^x\leq 1+2x$ and $(1-x)(1-y)\geq 1-x-y$ for $0\leq x,y\leq 1$. Let $X$ have density proportional to $\exp(-\frac{1}{c\sigma}\|X\|)$. The posterior likelihood ratio is given by

$$f(\hat{\theta})\overset{\text{def}}{=}\frac{\mathbb{P}(\mathcal{D}_{\text{train}}\in\mathbb{D}^{\text{in}}\mid\hat{\theta})}{\mathbb{P}(\mathcal{D}_{\text{train}}\in\mathbb{D}^{\text{out}}\mid\hat{\theta})}=\frac{\sum_{i=1}^N\exp(-\frac{1}{c\sigma}\|\hat{\theta}-a_i\|)}{\sum_{j=1}^N\exp(-\frac{1}{c\sigma}\|\hat{\theta}-b_j\|)}.$$

We claim that for all $\hat{\theta}$ with $\|\hat{\theta}\|\leq\gamma\sigma c\log c$, $1-\frac{\eta}{2}\leq f(\hat{\theta})\leq(1-\frac{\eta}{2})^{-1}$. First, suppose that $\|\hat{\theta}\|\leq c^\alpha\sigma$. Then we have:

$$f(\hat{\theta})\geq\frac{\sum_{\|a_i\|\leq c^\alpha\sigma}\exp[-\frac{1}{c\sigma}(\|\hat{\theta}\|+\|a_i\|)]}{N}$$

$$\geq\frac{(1-\frac{2}{c^{M\alpha}})N\cdot e^{-2c^{\alpha-1}}}{N}$$

$$\geq 1-4c^{-\min(M\alpha,1-\alpha)}. \tag{10}$$

Otherwise, $\|\hat{\theta}\| \geq c^{\alpha}\sigma$. We now have the following chain of inequalities:

$$f(\hat{\theta}) \geq \frac{\sum_{\|a_i\|\leq c^{\alpha}\sigma} e^{-\frac{1}{c\sigma}(\|\hat{\theta}\|+\|a_i\|)}}{\sum_{\|b_j\|\leq c^{\alpha}\sigma} e^{-\frac{1}{c\sigma}(\|\hat{\theta}\|-\|b_j\|)} + \sum_{c^{\alpha}\sigma<\|b_i\|<\|\hat{\theta}\|} e^{-\frac{1}{c\sigma}(\|\hat{\theta}\|-\|b_j\|)} + \sum_{\|b_i\|\geq\|\hat{\theta}\|} e^{-\frac{1}{c\sigma}(\|b_i\|-\|\hat{\theta}\|)}}$$

$$= \frac{\sum_{\|a_i\|\leq c^{\alpha}\sigma} e^{-\frac{1}{c\sigma}\|a_i\|}}{\sum_{\|b_j\|\leq c^{\alpha}\sigma} e^{\frac{1}{c\sigma}\|b_j\|} + \sum_{c^{\alpha}\sigma<\|b_i\|<\|\hat{\theta}\|} e^{\frac{1}{c\sigma}\|b_j\|} + \sum_{\|b_i\|\geq\|\hat{\theta}\|} e^{\frac{1}{c\sigma}(2\|\hat{\theta}\|-\|b_i\|)}}$$

$$\geq \frac{N(1-\frac{2}{c^{M\alpha}})e^{-c^{\alpha-1}}}{N\left(e^{c^{\alpha-1}} + \frac{2}{c^{M\alpha}}e^{\frac{1}{c\sigma}\|\hat{\theta}\|} + \frac{2\sigma M}{\|\hat{\theta}\|^M}e^{\frac{1}{c\sigma}\|\hat{\theta}\|}\right)}$$

$$\geq \frac{(1-\frac{2}{c^{M\alpha}})e^{-c^{\alpha-1}}}{e^{c^{\alpha-1}} + \frac{2}{c^{M\alpha}}e^{\gamma\log c} + \frac{2}{c^{M\alpha}}e^{\gamma\log c}}$$

$$\geq 1 - 2c^{-M\alpha} - c^{\alpha-1} - 2c^{\alpha-1} - 4c^{\gamma-M\alpha} \tag{11}$$

$$\geq 1 - 9c^{-\min(1-\alpha, M\alpha-2\gamma)}.$$

Combining this with (10) shows that $f(\hat{\theta}) \geq 1 - 9c^{-\min(1-\alpha, M\alpha-\gamma)}$ for all $\|\hat{\theta}\| \leq \gamma\sigma c\log c$.

Next, we must measure the probability of $\|\hat{\theta}\| \leq \gamma\sigma c\log c$. We can lower bound this probability by first conditioning on the value of $\mathcal{D}_{\text{train}}$:

$$\mathbb{P}(\|\hat{\theta}\| \leq \gamma\sigma c\log c) = \frac{1}{|\mathbb{D}|}\sum_{D\in\mathbb{D}} \mathbb{P}(\|\hat{\theta}\| \leq \gamma\sigma c\log c \mid \mathcal{D}_{\text{train}} = D)$$

$$\geq \frac{1}{|\mathbb{D}|}\sum_{\|\mathcal{A}(D)\|\leq c\sigma} \mathbb{P}(\|X\| \leq \gamma\sigma c\log c - \|\mathcal{A}(D)\|)$$

$$\geq \left(1-\frac{1}{c^M}\right)\left(1 - \frac{1}{2}\exp\left(-\frac{\gamma\sigma c\log c - c\sigma}{c\sigma}\right)\right)$$

$$= \left(1-\frac{1}{c^M}\right)\left(1 - \frac{e}{2}c^{-\gamma}\right)$$

$$\geq 1 - c^{-M} - \frac{e}{2}c^{-\gamma}.$$

Note that the exact same logic (reversing the roles of the $a_i$'s and $b_j$'s) shows that $f(\hat{\theta}) \leq (1 - 9c^{-\min(1-\alpha, M\alpha-2\gamma)})^{-1}$ with probability at least $1 - c^{-M} - \frac{e}{2}c^{-\gamma}$ as well.

Finally, we can invoke the result of Proposition 2. Let $\Delta = 9c^{-\min(1-\alpha, M\alpha-\gamma)}$ and note that $1 - \Delta \leq f(\hat{\theta}) \leq (1-\Delta)^{-1}$ implies that $\max\left\{\mathbb{P}(\mathbf{x}^* \in \mathcal{D}_{\text{train}} \mid \hat{\theta}), \mathbb{P}(\mathbf{x}^* \notin \mathcal{D}_{\text{train}} \mid \hat{\theta})\right\} \leq \frac{1}{2} + \frac{\Delta}{2}$.

Thus we have

$$\int \max\left\{\mathbb{P}(\mathbf{x}^* \in \mathcal{D}_{\text{train}} \mid \hat{\theta}), \mathbb{P}(\mathbf{x}^* \notin \mathcal{D}_{\text{train}} \mid \hat{\theta})\right\} d\mathbb{P}(\hat{\theta})$$

$$\leq \left(\frac{1}{2} + \frac{\Delta}{2}\right)\mathbb{P}(f(\hat{\theta}) \in [1 - \Delta, (1 - \Delta)^{-1}]) + \mathbb{P}(f(\hat{\theta}) \notin [1 - \Delta, (1 - \Delta)^{-1}])$$

$$\leq \frac{1}{2} + \frac{9}{2}c^{-\min(1-\alpha, M\alpha-\gamma)} + 2c^{-M} + ec^{-\gamma}$$

$$\leq \frac{1}{2} + \left(\frac{9}{2} + e\right)c^{-\min(1-\alpha, M\alpha-\gamma, \gamma)} + 2c^{-M} \tag{12}$$

$$\leq \frac{1}{2} + 7.5c^{-\frac{M}{M+2}}$$

where the last inequality follows by setting $\gamma = 1 - \alpha = M\alpha - \gamma$ and solving, yielding $\gamma = M/(M + 2)$. Solving for $\eta = 7.5c^{-\frac{M}{M+2}}$, we find that $c = (\frac{7.5}{\eta})^{1+2/M}$ suffices. This completes the proof. $\qquad\square$

**Corollary 11.** *When $M \geq 2$, taking $c = (6.16/\eta)^{1+2/M}$ guarantees $\eta$-MIP.*

*Proof.* The constant improves as $M$ increases, so it suffices to consider $M = 2$. Let $M = 2$ and $\alpha = \gamma = 1/2$, and refer to the proof of Theorem 7. Equation (11) can be improved to

$$1 - 2c^{-M\alpha} - c^{\alpha-1} - (e-1)c^{\alpha-1} - 4c^{\gamma-M\alpha} = 1 - (4 + e + 2c^{-1/2})c^{-1/2}$$

using the inequality $e^x \leq 1 + (e-1)x$ for $0 \leq x \leq 1$ instead of $e^x \leq 1 + 2x$, which was used to prove Theorem 7. With $\Delta = (4 + e + 2c^{-1/2})c^{-1/2}$, (12) becomes

$$\frac{1}{2} + \left(\frac{4 + e + 2c^{-1/2}}{2} + e + 2c^{-3/2}\right)c^{-1/2}. \tag{13}$$

Observe that since $\eta \leq 1/2$, when we set $c = (6.16/\eta)^2$, we always have $c \geq (2 \cdot 6.16)^2$, in which case

$$\frac{4 + e + 2c^{-1/2}}{2} + e + 2c^{-3/2} \leq 6.1597.$$

Thus, with $c = (6.16/\eta)^2$, we have

$$(13) \leq \frac{1}{2} + 6.1597 \cdot \frac{\eta}{6.16} \leq \frac{1}{2} + \eta.$$

This completes the proof. $\qquad\square$

**Corollary 8.** *Let $\sigma_i^M \geq \mathbb{E}|\theta_i - \mathbb{E}\theta_i|^M$, and define $\|x\|_{\sigma,M} = \left(\sum_{i=1}^d \frac{|x_i|^M}{d\sigma_i^M}\right)^{1/M}$. Generate*

$$Y_i \sim \text{GenNormal}(0, \sigma_i, M), \quad U = Y/\|Y\|_{\sigma,M}$$

*and draw $r \sim \text{Laplace}\left((6.16/\eta)^{1+2/M}\right)$. Finally, set $X = rU$ and return $\hat{\theta} = \theta + X$. Then $\hat{\theta}$ is $\eta$-MIP.*

*Proof.* We with to apply the result of Theorem 7 with $\|\cdot\| = \|\cdot\|_{\sigma,M}$. To do this, we must bound the resulting $\sigma^M$ and show that the density of $X$ has the correct form. First, observe that

$$\sigma^M = \mathbb{E}\|\theta - \mathbb{E}\theta\|^M = \sum_{i=1}^d \mathbb{E}\frac{|\theta_i - \mathbb{E}\theta_i|^M}{d\sigma_i^M} \leq \sum_{i=1}^d \frac{1}{d} = 1.$$

It remains to show that the density has the correct form, i.e. depends on $X$ only through $\|X\|$. This will be the case if the marginal density of $U$ is uniform. Let $p(U)$ be the density of $U$. Observe that,

for any $\|u\| = \|u\|_{s,M} = 1$, we have that $Y \mapsto u$ iff $Y = su$ for some $s > 0$. Thus

$$p(u) \propto \int_{s=0}^{\infty} e^{\frac{1}{\sigma_1^2}(su_1/\sigma_1)^M + \cdots + (su_d/\sigma_d)^M} ds$$

$$= \int_{s=0}^{\infty} e^{-s^M d\|u\|^2} ds$$

$$= \int_{s=0}^{\infty} e^{-s^M d} ds.$$

The last inequality holds because $\|u\| = 1$ is constant. Thus, the density is independent of $u$ and we can directly apply Theorem 7. □

**Proposition 9.** *For any finite $D \subseteq \mathbb{R}$, define $\mathcal{A}(D) = \frac{1}{\sum_{x \in D} x}$. Given a dataset $\mathcal{D}$ of size $n$, define $\mathbb{D} = \{D \subseteq \mathcal{D} : |D| = \lfloor n/2 \rfloor\}$, and define*

$$\sigma^2 = \mathrm{Var}(\mathcal{A}(D)), \qquad \Delta = \max_{D \sim D' \in \mathbb{D}} |\mathcal{A}(D) - \mathcal{A}(D')|.$$

*Here the variance is taken over $D \sim \mathrm{Unif}(\mathbb{D})$. Then for all $n$, there exists a dataset $|\mathcal{D}| = n$ such that $\sigma^2 = O(1)$ but $\Delta = \Omega(2^{n/3})$.*

*Proof sketch.* Assume $n$ is even for simplicity. Let $p = \binom{n}{n/2}^{-1}$ and $A = \sqrt{p} - \sum_{i=0}^{\frac{n}{2}-2} 2^i$. Take

$$\mathcal{D} = \{2^i : i = 0, \ldots, n-2\} \cup \{A\}.$$

When $D = \{2^0, \ldots, 2^{\frac{n}{2}-2}, A\}$, then $\mathcal{A}(D) = p^{-1/2}$, and this occurs with probability $p$. For all other subsets $D'$, $0 \leq \mathcal{A}(D') \leq 1$. □

**Lemma 12** (Chebyshev's Inequality)**.** *Let $\| \cdot \|$ be any norm and $X$ be a random vector with $\mathbb{E}\|X - \mathbb{E}X\|^2 \leq \sigma^k$. Then for any $t > 0$, we have*

$$\mathbb{P}(\|X - \mathbb{E}X\| > t\sigma) \leq 1/t^k.$$

*Proof.* This follows almost directly from Markov's inequality:

$$\mathbb{P}(\|X - \mathbb{E}X\| > t\sigma) = \mathbb{P}(\|X - \mathbb{E}X\|^k > t^k \sigma^k) \leq \frac{\mathbb{E}\|X - \mathbb{E}X\|^k}{t^k \sigma^k} \leq 1/t^k.$$

□

## C  Further Experiment Details & Discussion

**Details on Figure 1**  For DP, the value along the $x$-axis is given by the (tight) correspondence in Theorem 4, namely $\eta = \frac{1}{1+e^{-\varepsilon}} - \frac{1}{2}$. $\eta = 0$ corresponds to perfect privacy (the attacker cannot do any better than random guessing), while $\eta = \frac{1}{2}$ corresponds to no privacy (the attacker can determine membership with perfect accuracy). On the $y$-axis, for MIP, by Theorem 7, this is $(6.16/\eta)^2 \sigma$ where $\sigma$ is an upper bound on the variance of the base algorithm over random subsets, and for DP this is $\frac{\Delta}{\log \frac{1+2\eta}{1-2\eta}}$. (This comes from solving $\eta = \frac{1}{1+e^{-\varepsilon}} - \frac{1}{2}$ for $\varepsilon$, then using the fact that $\mathrm{Laplace}(\Delta/\varepsilon)$ noise must be added to guarantee $\varepsilon$-DP.)

**Considerations When Choosing $M$**  Figure 2 indicates that MIP continues to improve as $M$ increases. The value of $\sigma_i$ is just a centered $\ell_p$-norm with $p = M$, and for "reasonable" distributions, these do not grow too quickly as $M$ increases. In particular, their growth is offset by the decrease in $(6.16/\eta)^{1+2/M}$, leading to improved results. However, one must still be careful not to choose $M$ too large for several reasons. First, we certainly cannot take $M \to \infty$; in the limit, $\sigma_i \xrightarrow{M \to \infty} \max_{\mathcal{D}_{\mathrm{train}}} |\theta_i - \tilde{\theta}_i|$, which is a quantity that cannot be estimated from data without computing $\theta$ for *all* random train/holdout splits of the private data. The true value of $\sigma_i$ approaches this quantity for $M$ large but finite, and as such, much larger samples will be required to get an accurate estimate of $\sigma_i$ from data. In addition to these statistical concerns, there are also numerical issues. For instance, we found that for larger $M$ (e.g. $M = 10$), exponentiation by $M$ often left quantities either too close to 0 or too large to be accurately handled with finite numerical precision, leading to either no noise or far too much noise being added. Thus for practical purposes, we recommend using moderate values of $M$.

