# OpenReview forum: "Provable Membership Inference Privacy"
_NeurIPS.cc/2022/Workshop/TSRML — TSRML2022_

### Official Review · Reviewer_Qsuk · 2022-10-19
**Interesting contribution of a new privacy notion**

**Overall Rating:** 7

**Summary:**

The authors suggest a new privacy notion. The notion is based on the observation that many credible threads to the privacy of MLs model stem from membership inference attacks. The authors then suggest a definition of privacy they call “re-identification privacy” that is aimed at protecting models from membership inference attacks. Relative to the standard notion of differential privacy, the authors show that their privacy notion is less strict, however, potentially allowing for more favorable utility / privacy tradeoffs. Rudimentary experiments showcase the utility of the suggested privacy notion.

**Strengths:**

- The authors extensively discuss their suggested privacy notions and its usefulness relative to the standard DP notion. For example, similar to DP privacy notions, the authors show that a post processing quality holds for their notion.
- The authors also suggest a simple method to defend against membership inference, and demonstrate that substantially less noise is required to achieve their privacy notion relative to DP.
- A toy experiment highlights the effectiveness of the proposed privacy notion.


**Weaknesses:**

- An experimental evaluation showcasing the real-world utility of the authors suggested privacy notion is missing. I would suggest putting the suggested defense to a test on real world data; this would mean to include experimental results for various membership inference attacks including attacks on unsupervised (i.e., MI attacks on data generation) and supervised learning (i.e., MI attacks on classification or regression tasks) tasks, and comparing the utility/privacy tradeoffs by DP versus the ones by Re-identification privacy.
- The discussion on how RIP differs from DP and what implications that has for the construction of private algorithms could have been made more clear.


**Minor suggestions**
- Consider changing the name of your privacy notion: attacks against identification are more commonly known as “membership inference attacks”. Therefore, it could make sense to call it MI privacy.
- In line 170 you write Theorem “10”, but you mean Theorem 3
- Line 158: If A known, and random seed known, we are doomed: this is not immediately clear.
- How do your results differ from the technically closest work? To what extent do your results complement the results by Thudi et al (2022)? I would have liked to see a deeper discussion on this part.
- D_{synth} is the output of a model trained to output synthetically generated data I suppose, but this has not been previously defined.

**Overall Recommendation:**

I recommend acceptance. I think the manuscript provides many interesting insights, is overall easy to follow, and the connection of privacy notions to membership inference attacks is interesting. That being said, I am not an expert on differential privacy and I also did not check the correctness of the proofs.

**Review Confidence:**

3: The reviewer is fairly confident that the evaluation is correct

---

### Official Review · Reviewer_9nta · 2022-10-20
**A new definition to study privacy preservation of ML models**

**Overall Rating:** 4

**Summary:**

The paper proposes a new definition to study the statistical privacy preservation of ML models. The definition can be directly transformed into practical algorithms to estimate its privacy guarantees. Theoretical analysis is given where also connections to eps-DP guarantees are made.



**Strengths:**

- I think this is generally a nice idea and a timely topic. As stated in the abstract, DP guarantees are often too strong and difficult to interpret. And that the empirical risks of privacy leakage can be much
smaller than what is indicated by the guarantees.

- I think the presentation is clear, paper is well written.

**Weaknesses:**

- I think there is a one major deficit with the paper. Let me explain:

You write: "RIP guarantees are easily interpretable in terms of the success rate of membership inference attacks." You are neglecting the fact that so are DP guarantees, and you are also neglecting the work related to this fact, see e.g.

Nasr, Milad, et al. "Adversary instantiation: Lower bounds for differentially private machine learning." 2021 IEEE Symposium on Security and Privacy (SP). IEEE, 2021.

Kairouz, Peter, Sewoong Oh, and Pramod Viswanath. "The composition theorem for differential privacy." International conference on machine learning. PMLR, 2015.

Wasserman, L., & Zhou, S. (2010). A statistical framework for differential privacy. Journal of the American Statistical Association, 105(489), 375-389.

As you see, e.g. in Eq. (4) of Nasr et al. (2021), the False Negative and False Positive ratios of a given decision rule (e.g. decides based on output of a mechanism whether data element x is in D or not) can be directly linked to (eps,delta)-guarantees of the underlying mechanism. You can derive lower bounds for FNs and FPs via epsilons and deltas and vice versa: you can derive empirical epsilons with empirical FNs and FPs. I would also claim that also these bounds for FN/FP are tight (as your bounds for RIP) in a sense that one can easily construct examples (e.g. add Gaussian noise to a single binary data), where the relation in one-to-one.

I believe that the definition given here (Def 1) could be linked to these existing results. If you take k=n in your definition, i.e. D_train = D, then there seems to be almost one-to-one correspondence with the FN/FP-(eps,delta)-equivalence. And if you sample D_train uniformly randomly, you could use subsampling results to derive (eps,delta)'s for ´A(D_train), so that RIP could then be linked to FN/FP-(eps,delta)-equivalence.

So the connection to this FN/FP-(eps,delta)-equivalence is missing, and particularly since are quite strong statements such as (just an example, there are more) :

p.8:  "RIP allows us to add much less noise than what would be required by naively applying DP",

I think the connections should be studied more carefully.

**Overall Recommendation:**

Due to the deficit decribed above (i.e., the proposed new definition is not properly linked DP and existing results about DP, and also some crucial references are missing), I am leaning towards rejection. In any case, I think this topic would deserve more discussion.

**Review Confidence:**

4: The reviewer is confident but not absolutely certain that the evaluation is correct

---

### Decision · Program_Chairs · 2022-10-23

**Decision:**

Accept

**Comment:**

Reviewers have concerns on the soundness on the approach. I tend to accept this paper because it is highly relevant to the workshop, however please do address the concerns the reviewers had in the final version.